# Use of Newer and Repurposed Antibiotics against Gram-Negative Bacteria in Neonates

**DOI:** 10.3390/antibiotics12061072

**Published:** 2023-06-19

**Authors:** Angeliki Kontou, Maria Kourti, Elias Iosifidis, Kosmas Sarafidis, Emmanuel Roilides

**Affiliations:** 11st Department of Neonatology and Neonatal Intensive Care Unit, School of Medicine, Faculty of Health Sciences, Aristotle University of Thessaloniki, Hippokration General Hospital, Thessaloniki 54642, Greece; angiekon2001@yahoo.gr (A.K.); kosaraf@auth.gr (K.S.); 2Infectious Diseases Unit, 3rd Department of Pediatrics, School of Medicine, Faculty of Health Sciences, Aristotle University of Thessaloniki, Hippokration General Hospital, Thessaloniki 54642, Greece; makourti@icloud.com (M.K.); iosifidish@auth.gr (E.I.); 3Basic and Translational Research Unit, Special Unit for Biomedical Research and Education, School of Medicine, Faculty of Health Sciences, Aristotle University of Thessaloniki, Thessaloniki 54642, Greece

**Keywords:** neonates, ceftazidime–avibactam, ceftolozane/tazobactam, imipenem/cilastatin–relabactam, meropenem–vaborbactam, colistin, tigecycline, fosfomycin

## Abstract

Antimicrobial resistance has become a significant public health problem globally with multidrug resistant Gram negative (MDR-GN) bacteria being the main representatives. The emergence of these pathogens in neonatal settings threatens the well-being of the vulnerable neonatal population given the dearth of safe and effective therapeutic options. Evidence from studies mainly in adults is now available for several novel antimicrobial compounds, such as new β-lactam/β-lactamase inhibitors (e.g., ceftazidime–avibactam, meropenem–vaborbactam, imipenem/cilastatin–relebactam), although old antibiotics such as colistin, tigecycline, and fosfomycin are also encompassed in the fight against MDR-GN infections that remain challenging. Data in the neonatal population are scarce, with few clinical trials enrolling neonates for the evaluation of the efficacy, safety, and dosing of new antibiotics, while the majority of old antibiotics are used off-label. In this article we review data about some novel and old antibiotics that are active against MDR-GN bacteria causing sepsis and are of interest to be used in the neonatal population.

## 1. Introduction

Neonatal bacterial sepsis remains one of the major culprits of neonatal morbidity and mortality, especially in hospitalized term and preterm neonates all around the world and especially in low- and middle-income countries. An estimated 1.3 million episodes of neonatal sepsis occur annually with 200,000 sepsis-attributable deaths each year worldwide, while severe bacterial infections are responsible for approximately 3% of disability-adjusted life years (DALYs) in neonates [1,2,3].

Antimicrobial resistance (AMR) is a global public health threat; almost 5 million deaths in 2019 were associated with AMR affecting both high-income and low–middle-income countries, with the three most common pathogens with AMR being *Escherichia coli*, *Staphylococcus aureus*, and *Klebsiella pneumoniae* [3]. According to World Health Organization (WHO) priority list of non-mycobacterial antibiotic-resistant bacteria, carbapenem-resistant *Enterobacterales* (CRE) and third generation cephalosporin-resistant *Enterobacterales* (3GCRE) are of critical priority; whereas, methicillin-resistant *S. aureus* (MRSA) and vancomycin-resistant *Enterococcus* (VRE) are of high priority [4]. In several countries in the European region, with a north-to-south and west-to-east gradient, high percentages of resistance to third-generation cephalosporins and carbapenems in *K. pneumoniae* and high percentages of carbapenem-resistant *Acinetobacter* species and *Pseudomonas aeruginosa* are of significant concern [5]. A population-based modelling analysis using data from point prevalence European Centre for Disease Prevention and Control (ECDC) studies and surveillance data on AMR found an estimation of 33,110 attributable deaths and 874,541 DALYs due to healthcare-associated infections caused by antibiotic-resistant bacteria whose burden was highest in infants (<1 year old) and people older than 65 years; CREs, as well as other multidrug resistant organisms (MDROs) such as 3GCRE, MRSA, and VRE, were most frequent in infants [6].

Antimicrobial resistance for *Enterobacterales* is primarily based on the production of extended-spectrum β-lactamases (ESBLs) and carbapenemases. The production of these enzymes renders the current β-lactams ineffective against resistant Gram-negative bacteria (GNB). Resistance to carbapenems is complex and two mechanisms are mainly involved: (a) β-lactamase activity combined with structural mutations and (b) enzymes (carbapenemases) that hydrolyze carbapenem antibiotics. The former mechanism includes non-carbapenemase β-lactamases: ESBLs, generally encoded by plasmids, and AmpC cephalosporinases (AmpC), for which expression in *Enterobacterales* is most often associated with hyperproduction from inducible or derepressed chromosomal genes. ESBLs and AmpC confer carbapenem resistance along with the mutation of porins, a family of proteins of the outer membrane of Gram-negative bacteria that, when altered or lost, retard the diffusion of antibiotics across bacterial membrane to a rate slow enough to facilitate the action of ESBL and AmpC enzymes [7]. AmpC, for example, can bind carbapenems in the periplasm preventing them from accessing their targets, given that the enzymes are produced at a high level and the permeability of the outer membrane is reduced by the loss of porins [8]. Carbapenemases are classified by their molecular structures in three classes of β-lactamases according to the Ambler classification system: classes A, B, and D. Class A consists of serine carbapenemases mainly of the *K. pneumoniae*-producing carbapenemase (KPC) type. Class B are metallo-β-lactamases mainly of the Νew Delhi metallo-β-lactamase (NDM) type and of the Verona Integrated metallo-β-lactamase (VIM) type. Class D comprises oxacillinase-type carbapenemases, where OXA-48-like enzymes predominate [7].

The burden of neonatal late onset sepsis (LOS) due to MDR bacteria is exceptionally high in many regions of the world. AMR increase in the last decade has rendered most antibiotics of no utility. Resistance to even “WHO reserve” antibiotics has dramatically increased with 50–70% of the common Gram-negative clinical isolates now being MDR [9]. A large, multinational observational study showed that *K. pneumoniae*, *E. coli*, and *Enterobacter* spp. are the main Gram-negative bacteria responsible for neonatal sepsis with more than half of isolates being resistant to at least one antibiotic within four to six classes of antibiotics [10]. Data from positive blood cultures of hospitalized neonates in NICUs participating in the Neonatal AMR research network revealed carbapenem resistance rates of up to 84% [11]. Colonization rates with MDR are variable among NICUs; in a NICU in Ecuador, more than half of the neonates were colonized with ESBL-producing *Enterobacterales*, while colonization rates with CRO ranges from 1 to 25% [12,13,14]. Whether or not previous colonization with MDR is a significant risk factor for subsequent infection and the prognostic value of neonatal screening for the development of LOS need further clarification [12,15]. Moreover, higher mortality and morbidity is attributed to neonatal sepsis due to MDROs compared with non-MDROs, with case fatality rates of neonatal and pediatric sepsis due to CRO reaching 36% [16,17].

The limited therapeutic options against antimicrobial drug-resistant Gram-negative bacteria have led to the development and study of several novel antibacterial agents including β-lactam/β-lactamase inhibitor combinations (BL-BLIs) and the use of old or repurposed antibiotics. A framework for selecting appropriate therapy for children infected with CRE based on expert opinion has been proposed [18]. The Infectious Diseases Society of America (IDSA) annually updates their “suggested approaches” to the treatment of infections caused by extended-spectrum β-lactamase and AmpC β-lactamase producing *Enterobacterales* (ESBL-E), carbapenem-resistant *Enterobacterales* and *Acinetobacter baumannii*, *P. aeruginosa* with difficult-to-treat resistance (DTR-*P. aeruginosa*), and *Stenotrophomonas maltophilia*. These suggested approaches apply to both adult and pediatric populations, although there is a clear paucity of data on the treatment of such infections in children [17,19,20]. Not surprisingly, the above guidance reports are not addressed to neonates. According to recent systematic reviews about therapeutic options in neonates, the limited number of published articles, the low quality of evidence (retrospective data, heterogenous study design, and outcome definition, case series, or reports), and the very small sample size not permitting any statistical analysis further suggest that neonates remain “therapeutic orphans” in the fight against AMR [17,21]. A lack of evidence regarding pharmacokinetics (PK), safety, route (continuous or intermittent), dose and duration of administration, and the guidelines of using specific antibiotic(s) are the causes of the common off-label/unlicensed antibiotic use in neonates [22]. Despite the fact that, in the last decades (1998–2022), a significant increase in pediatric studies—submitted to the FDA—resulted in pediatric labeling changes, only six antimicrobials were labeled for use in neonates (linezolid, meropenem, ampicillin, ceftaroline fosamil, dalbavancin, clindamycin phosphate) (Figure 1) [23]. In this article, we review the data on some novel or repurposed antibiotics that are active against MDR Gram-negative (MDR-GN) bacteria causing sepsis and are of interest to be used in the neonatal population.

## 2. Novel β-Lactam-β-Lactamase Inhibitor (BL-BLI) Agents

### 2.1. Ceftazidime–Avibactam

Ceftazidime–avibactam (CAZ-AVI) is a newly developed antibiotic, one of the novel β-lactam agents combined with a β-lactamase inhibitor. Ceftazidime, a well-known broad spectrum third generation cephalosporin with antipseudomonal activity, is combined to avibactam, which is a new non-β-lactam β-lactamase inhibitor, able to inactivate several β-lactamases by forming a covalent adduct with the enzyme that is stable to hydrolysis. In this way, avibactam protects the degradation of ceftazidime allowing it to act against bacteria that would otherwise be resistant. In particular, avibactam inhibits Ambler class A (e.g., TEM-1, CTX-M-15, KPC-2, KPC-3), class C (e.g., AmpC), and certain class D β-lactamases (e.g., OXA-10, OXA-48), whereas it is inactive against Metallo-β-lactamases (class B enzymes e.g., NDM, VIM, IMP) [24,25]. Thus, CAZ-AVI is effective for the treatment of infections due to XDR *Enterobacterales* and *P. aeruginosa* when β-lactam resistance is due to the production of such β-lactamases. There are reports that the co-administration of CAZ-AVI and aztreonam can overcome resistance conferred by metallo-β-lactamases producing *Enterobacterales* and *P. aeruginosa* [26,27].

Before CAZ-AVI, the primary drug of choice for KPC infection was colistin, which has been known to have a severe side effect profile. Currently, CAZ-AVI is authorized in Europe for the treatment of complicated intra-abdominal infections (cIAIs), complicated urinary tract infections (cUTIs) including pyelonephritis, hospital-acquired pneumonia including ventilator-associated pneumonia (HAP/VAP), and infections due to aerobic MDR-GN bacteria susceptible to CAZ-AVI with limited or no other available therapeutic options in adults and children ≥3 months to <18 years old [28]. On the contrary, in the United States, CAZ-AVI has no approval for the treatment of HAP/VAP in pediatric patients ≥ 3 months to < 18 years old [29,30]. Meanwhile, in real clinical practice, CAZ-AVI is used off-label in the treatment of bloodstream infections (BSI), catheter-related bacteremia (CLABSI), endocarditis, osteomyelitis, ventriculitis, and mediastinitis; both observational and comparative studies focused on infections in adults due to KPC and OXA-48-producing *Enterobacterales* have shown promising results [31]. On the contrary, there is a paucity of data regarding treatment in pediatric patients with infections other than those approved, especially BSIs in critically ill children of all ages.

In healthy adult studies, both substances (ceftazidime and avibactam) show linear PK and share similar PK parameters allowing for their combined dosing. After intravenous administration, both agents have a half-life of nearly 2 h, exhibit low plasma protein binding (5–22.8% and 5–8.2%, respectively), and are not metabolized [31,32]. Renal clearance is the main route of elimination and the dose adjustment of CAZ-AVI is required in patients with moderate and severe renal impairment [28,33]. In pediatric patients of four age groups (group 1, ≥12 to <18 years; group 2, ≥6 to <12 years; group 3, ≥2 to <6 years; group 4, ≥3 months to <2 years) who received a single-dose of i.v. CAZ-AVI (group 1, 2000 to 500 mg; group 2, 2000 to 500 mg [>40 kg] or 50 to 12.5 mg/kg [<40 kg]; group 3 and 4, 50 to 12.5 mg/kg), the PK profiles of both ceftazidime and avibactam were comparable across the four age groups and broadly similar to those observed in adults who received a single dose of ceftazidime 2000 mg and avibactam 500 mg, administered intravenously over 2 h (Table 1) [34,35]. 

Furthermore, the updated combined adult and pediatric population PK models supported the approval of currently recommended pediatric dosage regimens for children with cIAI or cUTI and normal or mildly impaired renal function (creatinine clearance >50 mL/min/1.73 m^2^): ≥6 months to <18 years: 50/12.5 mg/kg (maximum 2000–500 mg); ≥3 to <6 months old: 40/10 mg/kg (every 8 h by 2-h intravenous infusion), which achieved exposures and a probability of target attainment comparable to those in adults [36]. Moreover, the administration of the same dosing regimens to children with HAP/VAP is supported [36].

At present, there are no PK data for neonates and infants <3 months, whereas there are scarce case reports on the safety and efficacy of CAZ-AVI in neonatal patients [37,38,39,40,41]. To the best of our knowledge, in the largest case series of eight pediatric patients, Iosifidis et al. reported the use of CAZ-AVI in five NICU preterm (GA: 25^+5d^–32^+4d^ weeks, PNA: 6–134 days, BW: 0.9–2 kg) and one early term neonate (GA: 37^+3d^ weeks, PNA: 21 d, BW: 2.4 kg) as empirical (2/6) or targeted (4/6) salvage therapy in combination with other antimicrobials, for probable or proven sepsis due to carbapenem-resistant *Klebsiella pneumoniae*; two preterm neonates were on septic shock. CAZ-AVI was administered intravenously (4–21 days) at 62.5 (50/12.5) mg/kg every 8 h, which is higher than the currently approved dose for infants 3 months of age. During CAZ-AVI therapy, two neonates developed hypomagnesemia, managed with an increased magnesium supplement in TPN, and one of them direct bilirubinemia, resolved 15 days later without significant intervention. As other antibiotics including colistin, fosfomycin, aminoglycosides, glycopeptides, and liposomal amphotericin B were co-administered, no clear causality to the drug could be attributed. No severe adverse events were reported and the outcome at 30 days was cure without relapse [41].

Similar increased doses were administered by Asfour et al. in two preterm neonates. The first case (BW: 920 g, GA: 27 wk, PNA: 3 wk) was treated with CAZ-AVI (21 d) combined with colistin (14 d) for *K. pneumoniae* BSI and meningitis; the second case was treated with CAZ-AVI (5 d) and amikacin (21 d) for *K. pneumoniae* BSI and, despite microbiological cure, the patient died at the fifth day of CAZ-AVI therapy, probably due to sepsis on the grounds of prematurity and chronic lung disease [38]. No other serious adverse events were observed, except a significant increase in creatinine of the second patient and, as dose adjustment is required in patients with renal impairment, the CAZ-AVI frequency was changed to every 24 h, although drug PK in neonates, especially in those with acute kidney injury, is unknown [38]. A 25-d old preemie (GA 27 wk) was successfully treated with a lower dose at 40/10 mg/kg/dose every 8 h, as targeted therapy for a UTI due to PDR *K. pneumoniae*. Glycosuria, which presented during treatment and spontaneously disappeared 5 days after the end of therapy, was referred as the only adverse event possibly related to the drug, due to the reversible impairment of renal tubular function [40]. In an ELBW (GA: 29 wks, BW: 890 gr) neonate, successfully treated for MDR *K. pneumoniae* bacteremia and AKI on peritoneal dialysis, the initial dose of 50 (40/10) mg/kg IV q8h was adjusted to 23.75 mg/kg i.v. q48h for 3 days, returning to the initial dose on the 5th day until the completion of a 14-day therapy, without reporting adverse events [39].

The off-label use of CAZ/AVI in a large number of neonates has been recently reported [42]. In this cohort, 21 neonates received 31 CAZ-AVI courses. The median gestational age at birth was 29 weeks and they had a median weight of 1170 g, and according to their APGAR, CRIB II, and SNAPPE scores, they had a medium/severe clinical status. The median postnatal age during the initiation of CAZ/AVI administration was 44 days. CAZ/AVI use was started empirically in more than half of cases at a dose of 20–50 mg/kg of ceftazidime every 8 h. The median treatment duration was 10 days but in most cases CAZ/AVI was co-administered with other antimicrobials (i.e., colistin, tigecycline, fosfomycin, amikacin). KPC producing *K. pneumoniae* was the most frequently isolated pathogen. However, there were three bloodstream infections due to XDR *A. baumannii*. Overall, clinical response was very good on day 15 and 30 (>74%). Five deaths were reported. However, all these neonates were critically ill with sepsis and treatment included antimicrobials with little or without safety data for the neonates (i.e., colistin, tigecycline, Fosfomycin, and daptomycin) and therefore potential adverse events associated with the use of CAZ/AVI cannot be drawn. For this reason, clinical trials of CAZ/AVI in premature neonates are warranted.

As CAZ-AVI may have a role in the treatment of neonates with serious infections due to XDR/MDR-GN bacteria, more clinical data on the use of the drug is an unmet need. A phase 2a, two-part, open-label, non-randomized, multicenter, single and multiple dose trial (ClinicalTrials.gov Identifier: NCT04126031), which just completed recruiting pediatric patients, aims to evaluate the pharmacokinetics, safety, and tolerability of single and multiple doses of intravenous CAZ/AVI in hospitalized infants and neonates from 26 weeks of gestation to 3 months of age with suspected or confirmed Gram-negative BSI [43]. According to the study protocol, CAZ/AVI is administered as a 2 h intravenous infusion at the following dosing regimens based on gestational, corrected, and postnatal age and on the current weight of the enrolled neonates: (i) 30/7.5 mg/kg/dose q12 (ceftazidime and avibactam, respectively) in the group of term infants (GA ≥ 37 weeks) with postnatal age > 28 days and preterm infants with corrected age >28 days to < 3 months old, (ii) 20/5 mg/kg/dose q12 in term neonates (GA ≥ 37 weeks) from birth to ≤ 28 days old, (iii) 20/5 mg/kg/dose q12 in the preterm neonates with GA ≥ 26 weeks to < 37 weeks from birth to ≤ 28 days old [43].

Several reports have addressed the emergence of resistance to CAZ/AVI beyond the intrinsic resistance of Gram-negative bacteria that harbor Ambler class B (metallo-β-lactamases) or some of the class D β-lactamases. In KPC-producing *K. pneumoniae* isolates, there have been several mutations (within or outside the omega loop region) that are associated with in vitro resistance to CAZ/AVI in patients with or without previous antimicrobial exposure [44]. In addition, the (over)expression of KPC in conjunction with other mechanisms of resistance such as porin mutations and other β-lactamases (VEB-25) has been documented in CAZ/AVI-resistant bacteria [45]. In a recent systematic review of clinical cases, CAZ/AVI resistant isolates were infrequently isolated all over the world, but their high fatality rate, as well as their rising rates, are of concern [46]. Meanwhile, besides resistance to ceftazidime/avibactam, the rapid identification of intrinsic resistance mechanisms such as Ambler class B carbapenemase is crucial for appropriate antibiotic selection and thus point-of-care testing and regional epidemiology surveillance are of great significance.

### 2.2. Ceftolozane/Tazobactam

Ceftolozane/tazobactam (C/T) is a combination of a semisynthetic, bactericidal, antipseudomonal, fifth generation cephalosporin, ceftolozane, with the known β-lactamase inhibitor, tazobactam. Ceftolozane inhibits bacterial cell wall biosynthesis through penicillin-binding proteins (PBPs); it has an enhanced affinity for the PBPs of *P. aeruginosa*, a high stability against Amp-C type β-lactamases, frequently present in *P. aeruginosa*; and it is significantly less affected by the changes in the porin permeability or efflux pumps of the external membrane of Gram-negative bacteria [47,48,49]. C/T has a broad coverage against Gram-negative organisms, particularly MDR and XDR *P. aeruginosa*, ESBL-producing *Enterobacterales*, and some anaerobes (*Bacteroides fragilis* and non-*Bacteroides* Gram-negatives) and some *Streptococcus* spp. (excluding Enterococcus), while it shows limited activity against ESBL-producing *K. pneumoniae*, carbapenemase-producing *Enterobacterales*, and anaerobic Gram-positive cocci [50].

C/T has been approved by the FDA since 2014 for complicated intra-abdominal infections (IAIs) combined with metronidazole and for complicated urinary tract infections (cUTIs) in adults (>18 years old) [51]. This indication was extended to HAP/VAP in 2019 [52]. On the other hand, in Europe, the drug is currently indicated for the treatment of cIAIs and cUTIs in pediatric patients and neonates of GA > 32 wks from the seventh day of life up to 18 years old, at 20 mg/kg ceftolozane/10 mg/kg tazobactam (up to a maximum dose of 1 g ceftolozane/0.5 g tazobactam) [53].

In a phase 1 open-label, single dose, multicenter study, seven neonates and young infants of GA > 32 wks and PNA 7 d to <3 months, and six neonates, of GA ≤ 32 weeks and PNA 7 days to < 3 months, with suspected/proven Gram-negative infection received 20/10 mg/kg and 20/10 mg/kg of an estimated glomerular filtration rate (eGFR) > 50 mL/min/1.73 m^2^ or 12/6 mg/kg if eGFR < 50 mL/min/1.73 m^2^, respectively. The PK profiles were generally comparable to those of older children but not surprisingly with greater interindividual variability, higher terminal half-lives, probably due to an increase in the volume of distribution, and decreased clearance, which are typical of neonates compared with older patients (Table 2). The drug was well tolerated without any serious adverse events [54].

A more recent phase 2, randomized trial studied the safety and efficacy of C/T vs. meropenem in 20 full-term neonates and young infants <3 months of age with pyelonephritis. C/T had a favorable safety profile in these patients, and the rates of clinical cure and microbiologic eradication were similar to meropenem [56]. More data on efficacy in premature neonates are needed.

### 2.3. Imipenem/Cilastatin–Relabactam

In an effort to restore the clinical activity of imipenem, relebactam, which is a novel β-lactamase inhibitor, was combined with imipenem/cilastatin, (an established anti-pseudomonal carbapenem). Relebactam exhibits a dual Ambler class A/C activity but confers no activity against class D OXA-48 and class B MBL producing *Enterobacterales* and carbapenem-resistant *A. baumannii* [57]. Imipenem/cilastatin–relebactam (IMI-REL) is indicated for patients over 18 years of age for the treatment of HABP/VABP due to susceptible Gram-negative bacteria and for complicated cUTIs and cIAIs with limited or no alternative treatment options [58]. In adults, phase 2 clinical trials have shown that IMI-REL is noninferior to imipenem/cilastatin in the treatment of cUTIs, including pyelonephritis, and cIAIs with comparable adverse reactions. The ongoing MK-7655A-016 phase 3 multinational, randomized clinical study (NCT03583333) is designed to evaluate the safety, tolerability, and efficacy of IMI-REL versus piperacillin/tazobactam in adult participants with HABP or VABP [59]. Another small phase 3 clinical trial has shown that IMI-REL is an efficacious and well-tolerated option compared with imipenem/cilastatin plus colistin for the treatment of HABP/VABP, cIAIs, and cUTIs caused by imipenem-non susceptible (but IMI-REL and colistin-susceptible) Gram-negative organisms with significantly reduced nephrotoxicity compared with imipenem/cilastatin plus colistin [60]. A recently completed pediatric clinical study (MK-7655A-020) showed that IMI-REL exhibited approximately dose-proportional PK and a single dose was generally well tolerated [61]. The ongoing MK-7655A-021 phase 2/3 open-label, randomized clinical study (NCT03969901) will provide valuable information for the pediatric and neonatal population with confirmed or suspected Gram-negative bacterial infection involving one of three primary infection types (HABP/VABP, cIAI or cUTI) [62].

### 2.4. Meropenem–Vaborbactam

Meropenem–vaborbactam (M/V) is a carbapenem β-lactamase inhibitor combination with activity against broad-spectrum β-lactamases in CRE infections. Vaborbactam, a cyclic boronic acid derivative, is a β-lactamase inhibitor with no antibacterial activity [63]. It prevents β-lactamases from hydrolyzing meropenem, which can then exert their action by disrupting bacterial cell-wall synthesis resulting in cell death. M/V shows a potent activity against class A carbapenemases (e.g., KPC-2, KPC-3, KPC-4, BKC-1, FRI-1, SME-2, NMC-A), class A ESBLs (CTX-M, TEM, SHV), and class C β-lactamases (CMY, P99, MIR, FOX) but not against metallo-β-lactamases (e.g., NDM, VIM, and IMP) and some class D carbapenemases (OXA-49-like) [63,64,65]. Therefore, M/V is mainly active against *Enterobacterales* with a KPC-mediated mechanism, but it has been shown that its activity is attenuated in isolates with a lack of ompK35 and ompK36 genes responsible for the encoding of outer membrane porins K35 and K36, respectively [63]. Moreover, M/V has been found to be active against strains producing KPC mutants with resistance to ceftazidime–avibactam (e.g., KPC-8, KPC-31), whereas vaborbactam does not protect meropenem hydrolysis against CR *Acinetobacter* spp. and *P. aeruginosa*, as meropenem resistance is largely attributed to mechanisms unrelated to the vaborbactam mode of action, such as outer-membrane impermeability, the upregulation of efflux systems, and the production of class B or class D β-lactamases [65,66,67]. The drug was first approved in the USA (FDA, August 2017) for the treatment of cUTI including pyelonephritis caused by susceptible *Escherichia coli*, *K. pneumoniae*, *and Enterobacter cloacae* species complex, while in Europe (EMA approval, November 2018), it is also indicated for the treatment of cIAI, hospital-acquired pneumonia (HAP), including ventilator-associated pneumonia (VAP) only in adult patients (≥18 years), at a dose regimen of 2 g/2 g every 8 h, as a 3 h intravenous infusion, for patients with normal renal function [68,69]. EUCAST provided a susceptibility clinical breakpoint of 8 mg/L for *Enterobacterales* and *P. aeruginosa*, while CLSI provided a susceptibility clinical breakpoint of 4 mg/L only for *Enterobacterales* [70].

Until now, PK, safety, and efficacy data derive from adult-only studies. To our knowledge, pediatric experience is limited to two case reports. Based on pharmacokinetic data of meropenem in critically ill children, Harnetty et al. administered a meropenem component of M/V at the dose of 40 mg/kg/dose every 6 h infused over 3 h, in a 4-year-old child with KPC *K. pneumoniae* bacteremia, which was successfully treated for 14 days. The dosing regimen provided a target attainment of 100% for meropenem serum concentrations above the minimum inhibitory concentration (MIC) for at least 40% of the dosing interval and was well tolerated [71]. In a 10-year-old cystic fibrosis female patient, infected with a PDR *Achromobacter* spp., meropenem–vaborbactam was co-administered (2 g, every 8 h, infused over 3 h) with cefiderocol and bacteriophage for 14 days; the combination was reported to be safe, effective, and well-tolerated [72]. An open label, phase 1 study evaluating the dosing, pharmacokinetics, safety, and tolerability of a single dose infusion of meropenem–vaborbactam in pediatric patients, from birth to less than 18 years of age with serious bacterial infections in stable condition (TANGOKIDS, ClinicalTrials.gov Identifier: NCT02687906) is currently being conducted and is still recruiting patients [73]. According to the study protocol, enrolled children of 12 to <18 years old received 40 mg/kg meropenem-40 mg/kg vaborbactam (2 g meropenem-2 g vaborbactam for subjects ≥50 kg), while after the analysis of the PK, safety, and tolerability data in this age group, the dose for ages 2 to <6 years was modified to 60 mg/kg (2 g meropenem-2 g vaborbactam for children weighting >33 kg) [73].

There is no published research on meropenem–vaborbactam use in neonates. On the contrary, meropenem, which has been approved by the FDA in infants < 3 months with complicated intra-abdominal infections since 2014, has been studied in both ill, hospitalized term and preterm neonates with LOS in a large multicenter phase III superiority RCT [74,75]. In terms of efficacy, a Neomero-1 trial showed that meropenem was not superior to SOC (ampicillin + gentamycin or cefuroxime + gentamycin), but the drug should be preferred in NICUs where LOS by ESBL and AmpC type beta-lactamases producing Gram-negative bacteria are common [75]. Neomero PK data and simulations showed that, in cases of increased MIC (up to 4 mg/L), doses should be increased to 40 mg/kg every 8 h to achieve therapeutic targets, and that longer infusions (up to continuous infusion) may increase plasma concentrations improving T > MIC, but worsen CSF penetration decreasing CSF T > MIC [76]. In a recently published PBPK study, using the target of 50% T > MIC for pathogens with MIC of 4 mg/L or 75% T > MIC for MIC of 2 mg/L, favorable target attainment was achieved across all dosing groups, further supporting the dosing regimen currently recommended by FDA [77].

## 3. Cefiderocol

Cefiderocol is a novel siderophore cephalosporin with a distinct mode of action and a broad spectrum of Gram-negative activity against difficult-to-treat bacterial infections [78]. A catechol moiety on the 3-position side chain attached to the cephalosporin molecule chelates free iron facilitating the entry of the cefiderocol–iron complex in the periplasmic space via active iron transporters. After dissociation from the iron, the drug binds to penicillin-binding proteins (PBPs), inhibiting synthesis of the bacterial cell wall [79]. Cefiderocol is stable against Ambler classes A, B, C, and D β-lactamases, with broad activity against GNB, including carbapenem-resistant *Enterobacterales* (CRE), *P. aeruginosa*, *Acinetobacter* spp., and *S. maltophilia* [78,80]. Its unique chemical structure and mode of action may offer protection against the loss of porin channels, the overexpression of efflux pumps, and the inactivation by carbapenemases [80]. In a large 2020 collection of Gram-negative isolates from hospitalized patients in USA and Europe, *Enterobacterales* susceptibility to cefiderocol was 99.8% and CRE susceptibility was 98.2%. Cefiderocol was the most active antimicrobial against all P. aeruginosa isolates (99.6% susceptible), whereas the susceptibility of *Acinetobacter* to cefiderocol was 97.7% and *S. maltophilia* was 100.0% (CLSI, 2021) and 97.9% (CLSI, 2022) [81]. According to a recent systematic review, β-New Delhi Metallo-β-lactamase positive isolates showed significantly higher MICs than other carbapenemase-producing *Enterobacterales* with a cefiderocol susceptibility rate of 83.4%, while the emergence of resistance has already been reported [82].

So far, based on three randomized controlled trials (RCTs) in adults—one phase II (APEKS-cUTI) and two phase III (APEKS-NP and CREDIBLE-CR)—examining its efficacy, cefiderocol has been FDA and EMA approved for treatment of cUTIs, HAP/VAP, and aerobic GN infections with limited therapeutic options only in adults, respectively [83,84,85,86]. It is administered intravenously as 2 g every 8 h as a 3 h infusion in adult patients with an estimated CrCl of 60–119 mL/min. It exhibits linear PK and since it is mainly eliminated via the kidneys, renal impairment alters the PK parameters as AUC, CL, t1/2 without significantly affecting Cmax and dosing regimens are modified according to renal function [82]. Dose adjustments are recommended for patients with renal impairment or augmented renal clearance [86]. According to the available evidence and expert opinion, cefiderocol could be used (i) as a preferred therapy for MBL-producing infections and less-common non-fermenters, (ii) as a reasonable alternative when b-lactam-b-lactamase inhibitors cannot be used for CR-*P. aeruginosa* infections, and (iii) as salvage therapy for CRAB infections in settings precluding the use of other agents [80].

The pharmacokinetics, safety, and efficacy of the drug in pediatric patients and neonates have not been established yet. Based on PK data from adults, a pharmacokinetic modeling study predicted PK in pediatric patients and proposed a pediatric dose regimen (Gestational age, GA < 32 weeks: 30 mg/kg for chronological age < 2 months, 40 mg/kg for 2 to < 3 months and 60 mg/kg for 3 months to < 18 years old with body weight < 34 kg; GA ≥ 32 weeks: 40 mg/kg for chronological age < 2 months, 60 mg/kg for 2 to < 3 months; and 2 gr for 3 months to < 18 years old with body weight ≥ 34 kg), which provides exposures comparable to adults. The proposed dose provided PTA > 90% for 75% fT > MIC against carbapenem non susceptible pathogens with MICs up to 4 μg/mL [87]. The safety, tolerability, and pharmacokinetics of cefiderocol in pediatric patients 3 months to < 18 years old are currently being investigated in two phase 2 clinical trials (ClinicalTrials.gov Identifier: NCT04215991 and NCT04335539) of single and multiple doses and the results are still pending [88,89].

The published experience in use of these drugs in neonates is limited to two cases of preterm neonates [90,91]. A preterm female neonate of GA: 31 weeks and a postnatal age of 20 days with a bloodstream infection due to a VIM-producing Klebsiella pneumoniae was successfully treated with cefiderocol for 9 days at the following dose regimen: loading dose 60 mg/kg and then 40 mg/kg every 8 h in extended infusion. The drug was well tolerated with rapid clinical and biochemical improvement while microbiological cure was recorded at 48 h of therapy [90]. Another preterm neonate of GA = 27, BW = 1040 gr after the ninth day of life presented with LOS due to CR *K. pneumoniae* only susceptible to colistin (MIC of < 0.5 by BMD) with NDM and OXA-48-like enzymes. Cefiderocol (30 mg/kg/ dose Q6H) was added to the initial antimicrobial treatment [meropenem + colistin + ceftazidime/avibactam + polymyxin B (after discontinuation of colistin due to creatinine derangement) + aztreonam] as a salvage therapy because of clinical deterioration, persistent thrombocytopenia, and positive blood culture even after 7 days of treatment. The neonate improved, platelets normalized, and blood cultures became negative. Cefiderocol was continued in combination with polymyxin B, CAZ-AVI, and AZT for 14 days with no adverse events reported [91].

## 4. Other Novel or Repurposed Antibacterial Agents

### 4.1. Colistin

Colistin is among the very few agents still effective against carbapenem-resistant Gram-negative bacteria. It has been used for clinical use since the late 1950s but was substituted some decades later by newer antimicrobials owing to reported neurotoxicity and nephrotoxicity. Recently, due to the stagnation of antibiotic development, colistin was re-evaluated as a last resort. It is a concentration-dependent antibiotic of the polymyxin class that is administered as the inactive form of colistimethate sodium (CMS), which is subsequently converted to the active form by the hydrolysis of methane sulphonate radicals [92]. Colistin also binds to endotoxins, thus reducing the release of inflammatory cytokines and blocking some of their biologic activity [93]. There is a paucity of PK data in pediatrics and neonates due to complicated pharmacokinetics, high interpatient variability, and narrow therapeutic indices. Therefore, recommendations for dosage in neonates are challenging. According to a PK study in neonates with normal renal function, the daily dose of CMS should be >150,000 IU/kg/day to achieve an average steady-state plasma colistin concentration (Css, avg) of >1 μg/mL, with close monitoring of renal function [94]. In addition, a recent PK study in critically ill children, including infants aged at least 1 month, found that colistin doses higher than those recommended by both EMA and FDA were associated with better antimicrobial exposure and without any additional safety concerns [95]. On the other hand, real life data from two global network databases that collected antibiotic prescribing data in children and neonates from hospitals around the world showed that almost 60% of neonates received colistin doses that were lower than those recommended by both FDA and EMA [96].

In neonates, the inhalational route is also used for the treatment of pneumonia and was first reported in 2010. Nebulized colistin as monotherapy was successfully administered and reported in neonates, but studies are scarce that support this as routine practice [97]. Moreover, it is suggested to use it in combination with intravenous colistin, since nebulized colistin alone might not reach the lung segments with pneumonia and a parenchymal loss of aeration. Intraventricular (IVT) CMS is used, and microbiological cure is reported in neonates and infants with meningitis in a dose ranging from 20,000 to 125,000 IU/kg/day. CMS and colistin cross the blood–brain barrier poorly despite the inflammation of meninges. Therefore, it is suggested to treat cerebrospinal infections with a combination of intravenous colistin with IVT or intrathecal CMS [98].

Colistin has been recommended by the Infectious Diseases Society of America (IDSA) 2022 guidance for the treatment of MDR-GN bacteria only as an alternate strategy when first-line options are not available or tolerated (mainly combination beta-lactamase inhibitors, carbapenems, or monobactam). Nevertheless, these recommendations are tailored for adults and high-income settings [19]. In low–middle-income countries (LMICs), colistin is the most prevalent antimicrobial and most studies come from these countries. Because of the concomitant use of other antimicrobials with colistin, the severity of the clinical condition in neonates, the lack of case control studies in neonates and infants, and the retrospective character of the studies reported, it is difficult to draw conclusions on the efficacy and safety of colistin in neonates. Renal impairment and electrolyte deficiencies, such as magnesium and potassium possibly related to renal tubulopathy, have been reported in a review of colistin use in neonates [99]. However, concomitant nephrotoxic agents and clinical comorbidities contribute to nephrotoxicity and renal injury. Moreover, evidence of neurotoxicity related to colistin use is rare in neonates. Nevertheless, prospective studies to evaluate the effect of colistin on the developing brain would be useful.

In conclusion, colistin appears to be a last resort agent in the fight against MDR-GN infections and its rational use is essential. It is suggested that the combination of colistin with other antibiotics can minimize the potential for the emergence of resistance with colistin monotherapy against *A. baumannii.* However, the optimal combination remains to be elucidated.

### 4.2. Tigecycline

Tigecycline is a bacteriostatic antimicrobial agent of the class of glycylcyclines (semi synthetic derivative of minocycline) with similarities to the structure and mechanism of the action of tetracyclines [100]. It exerts its action by binding to the bacterial 30S ribosome, blocking the entry of transfer RNA, which inhibits protein synthesis and bacterial growth, against a broad spectrum of Gram-positive and Gram-negative, anaerobic and atypical pathogens, including MDR and XDR microbes, such as MRSA, VRE spp., *A. baumannii*, and Gram-negative bacterial strains that produce ESBL and carbapenemases, with the exception of *Pseudomonas* spp. [101,102,103]. In the era of CRE, tigecycline is one of the last therapeutic options against infections due to such bacteria and its use in pediatric and neonatal populations is challenging as it is off-label [104,105].

Tigecycline is approved by the FDA for the intravenous treatment of cIAI, complicated skin and skin structure infections (cSSTI), and community acquired pneumonia (CAP). However, it is frequently used off-label for the treatment of HAP/VAP, rescue therapy for infections due to MDR bacteria, nosocomial urinary tract infections, and refractory *Clostridium difficile* infection [100,106,107]. FDA has approved its use only in ≥ 18 years old patients, at a loading dose of 100 mg and a maintenance dose of 50 mg twice daily, and warns that tigecycline should be preserved only if alternative antibacterial drugs are not available because of an observed increase in all-cause mortality in tigecycline-treated adult patients in a meta-analysis of 13 phase 3 and 4 clinical trials [107]. Meanwhile, the EMA has approved its use for ≥ 8 years old patients for the treatment of cIAI and cSSTI in situations where other alternative antibiotics are not suitable, providing a consultation with an infectious disease expert takes place, at 1.2 mg/kg q12h to a maximum dose of 50 mg q12h for 8–11 years old children, and at 50 mg q12h for 12–17 years old adolescents without the use of a loading dose [106]. Tigecycline pharmacokinetic properties, optimal dosing regimens, and the efficacy and safety data come mainly from studies in adults; published data for the pediatric population are limited to case series, case reports, and one open-label, phase 2, multiple ascending dose study, whereas for neonates, data are scarce [108,109,110,111]. The drug exhibits linear pharmacokinetics and its major routes of elimination include the excretion of unchanged drug into feces (through bile, 59%) and urine (renal, 33%), metabolic elimination (through glucuronidation and amide hydrolysis), and non-enzymatic degradation [112]. It has a long elimination t_1⁄2_ 37 ± 12 h and a large volume of distribution (9–10 L/kg) at a steady state, while it is bound to plasma proteins to a significant degree (71–87%), showing an atypical nonlinear protein binding [112,113]. Because of the high volume of distribution, the drug is rapidly accumulated in various tissue compartments, with a higher degree of penetration in the bile, gallbladder, and colon, and, to a lesser degree, in the lungs (even less so in the cerebral spinal fluid, synovial fluid, and bone), resulting in low bactericidal concentrations in the serum and epithelial lining fluid [114]. The latter findings offer a plausible explanation to the reported failures in tigecycline-treated adult patients with bloodstream infections and VAP at standard doses [115]. Several studies and expert opinions support the use of higher doses of tigecycline (100 mg twice daily) in adults and 2–3.2 mg/kg/dose q12 (after a loading dose) in children for treating CRE infections (e.g., HAP/VAP), especially from MBL-producing isolates, while combination therapy with other antibiotics is also suggested for bloodstream infections in severely ill patients [116,117,118]. These peculiar PK characteristics make the off-label use of tigecycline in neonates quite intimidating given the unique physiological and maturational characteristics of the neonatal population and the fact that bacteremia is the predominant type of neonatal infection.

To our knowledge, there are no PK data at all for infants or children < 8 years old, whereas in older children, doses higher than those currently proposed have not been studied in RCTs [110]. There are reports for the off-label use of a loading dose (1.8 mg/kg–6.5 mg/kg) and higher maintenance doses (1.25 mg/kg–3.2 mg/kg) in children (2.5 months to 14 years old), which were considered tolerable without serious adverse events [109,110,119]. The findings regarding efficacy and safety were similar to those observed in adults, although as tigecycline was administered in combination with other antibiotics, definite conclusions on the efficacy and safety of the drug in severe MDR/XDR infections in children cannot be drawn. There are scarce reports for the use of tigecycline in neonates [108,109,110]. Due to unavailable alternative treatments, Ipek et al. administered tigecycline to four critically ill preterm neonates as salvage combination therapy for the treatment of XDR *K. pneumoniae* BSI. Standard doses were administered, while in one neonate the dose was increased to 2 mg/kg q12 due to the persistence of bacteremia after 96 h of therapy. The outcome of all patients was favorable without serious adverse events. Interestingly, after the first week of treatment, all neonates presented with thrombocytopenia related to the drug, with PLT gradually returning to normal values after the end of therapy [120].

The liver function, hematology and coagulation parameters, amylase, and lipase should be monitored prior to the start and regularly during therapy [106]. It is generally considered that its use in children < 8 years old should be avoided due to the lack of safety data and to the potential adverse event of permanent tοοth discoloration, hitherto inadequately studied and confirmed. In a case-series of pediatric patients < 8 years old, the yellow staining of permanent teeth was presented in two out of twelve (17%) children who had received tigecycline at doses close to the recommended regimen for 8-11 years old for at least 19 days [121]. To clarify the efficacy, safety, and the optimal dosing regimen of tigecycline in neonates, well designed studies adjusted to their unique developmental physiological characteristics are needed.

### 4.3. Fosfomycin

Fosfomycin, a phosphoenolpyruvate (PEP) analogue, has recently been identified by WHO as a “critically important antimicrobial” [122]. It possesses a distinctive mechanism of bactericidal action by permanently inhibiting the primary step in the biosynthesis of peptidoglycan for bacterial wall synthesis [123]. It exhibits bactericidal activity against Gram-positive and Gram-negative pathogens including MRSA, VRE, CPE, and *P. aeruginosa* and may also penetrate biofilms [124]. Nevertheless, species naturally resistant to fosfomycin include *A. baumannii*, *S. maltophilia*, *Staphylococcus capitis*, *Staphylococcus saprophyticus*, *Mycobacterium tuberculosis*, *Vibrio sheri*, and *Chlamydia trachomatis* [125]. Fosfomycin’s unique mechanism of action permits synergy with other antibiotics (carbapenems and aminoglycosides), as has been demonstrated in vitro. Resistance to fosfomycin can develop rapidly when it is used in monotherapy and can be either chromosomal or plasmid-mediated [126].

The clinical efficacy of fosfomycin is well-documented in adults, especially for MDR urinary tract infections. Moreover, fosfomycin has been administered as a last-resort antibiotic choice for MDR pathogens in critically ill patients, especially in combination with other antibiotics with high clinical cure rates [127]. In the pediatric population, it is rarely administered and only occasionally prescribed for empirical use. There is limited existing literature describing the use of fosfomycin in neonatal sepsis. A series of studies have recently been published, which acknowledge its promising in vitro activity [128]. The potential utility of the combination of fosfomycin and amikacin for neonatal sepsis has been studied by assessing the in vitro activity and the nature and extent of any PD interactions and a candidate combination regimen suitable for further clinical study has been defined. An analysis of 247 Gram-negative bacteremia isolates from children revealed a high susceptibility rate among both *Enterobacterales* and *Pseudomonas* spp., including MDR and ESBL-producing organisms, in both community- and hospital-acquired infections and across both neonates and older children, rendering fosfomycin combined with aminoglycosides a new carbapenem-sparing regimen to treat antimicrobial-resistant neonatal and pediatric sepsis [129]. The recently published results of the NeoFosfo study [a single-centre open-label randomized controlled trial of 120 neonates aged ≤ 28 days treated with standard-of- care (SOC) antibiotics for sepsis: ampicillin and gentamicin, and half the participants were randomly assigned to receive additional intravenous and then oral fosfomycin at 100 mg/kg two times per day for up to 7 days (SOC-F), followed up for 28 days] suggest that an intravenous dose of 150 mg/kg two times per day is required for pharmacodynamic target attainment in most children, reduced to 100 mg/kg two times per day in neonates aged < 7 days or weighing < 1500 g [130]. Furthermore, intravenous and oral fosfomycin showed no evidence of impact on serum sodium or gastrointestinal side effects at 100 mg/kg two times per day, respectively [130]. Therefore, emerging evidence supports the validity of combination fosfomycin therapy as a promising life-saving last-resort antibacterial option for the treatment of neonatal sepsis caused by MDR bacteria. According to the results on iv and oral fosfomycin PKs in neonates with suspected clinical sepsis as part of the NeoFosfo study, estimates for total plasma clearance (CL): (8.94 L/h/70 kg) and the volume of the central compartment (Vc): (19.11 L/70 kg) were in agreement, with PK observed in healthy adults as is the calculated b phase half-life of 2.3 h (after 1 g of IV fosfomycin to healthy adults, V = 29.7 ± 5.7 L, CL = 8.7 ± 1.7 L/h, weight = 70.5 ± 11.1 kg, t1 = 2 = 2.4 ± 0.4 h). This is the first study to describe model-based oral bioavailability from cross-over data in a neonatal antibiotic study and the first report of neonatal fosfomycin CSF penetration. A two-compartment disposition model, with an additional CSF compartment and first-order absorption, was used and it best described the data. Bioavailability was estimated as 0.48 (95% CI = 0.347–0.775) and the CSF/plasma ratio as 0.32 (95% CI = 0.272–0.409). They developed a population pharmacokinetic model that could be used along currently available pharmacodynamic targets to select a neonatal fosfomycin dose based on an infant’s post menstrual age (PMA), postnatal age (PNA), and weight [131]. More solid data on dosing regimens, safety profiles, and appropriate combinations are needed before clear conclusions can be reached. Fosfomycin’s future place is still under evaluation, but it is probably as a companion drug to other IV antibiotics for difficult to treat infections, in variant dosing regimens.

## 5. Conclusions

Unfortunately, multidrug resistant organisms, especially Gram-negative bacteria, have entered NICUs and remain there, threatening the well-being of the most vulnerable neonatal population. In real practice, there is a great variability in antibiotic regimens used in neonates with clinicians often preferring the administration of combined regimens of two or more antibiotics [17]. According to a recent systematic review aiming to identify the current antimicrobial treatment options for MDR and XDR GNB infections in the neonatal population, colistin in combination with meropenem, amikacin, ciprofloxacin, or tigecycline is used for CRE neonatal infections, whereas, in association with other antimicrobials such as ciprofloxacin, it is prescribed for DTR and XDR *P. aeruginosa.* Furthermore, the most active antimicrobial for XDR *A. baumannii* seems to be colistin, whereas novel antimicrobials such as ceftazidime–avibactam are infrequently used as salvage therapy [21]. Novel antimicrobials seem to be promising based on experience from studies in adults and lately from a very small but increasing number of trials including neonates. Neonatologists face the problem of using many off-label antimicrobial agents and receiving a high volume of information regarding newer data of PK and safety even for old antibiotics. Moreover, pharmacologists or infectious diseases experts are not available in many NICUs. These problems make decision-making difficult. The treatment of neonatal sepsis due to MDR-GN bacteria is complex and challenging. Ideally, therapeutic decisions require expert consultation and an individualized approach until more evidence is available.

## 6. Future Directions

The battle against MDROs has to focus on two major fields: prevention and management. Prevention is mostly achieved by good infection control practices. However, often this is not perfect and MDR GN bacteria causes infections in the NICU; the off-label use of newly developed antimicrobials as well as the use of old antibiotics (not adequately studied and with dosing and safety concerns in the neonatal population) is a common practice in the NICU. Initiatives for the participation of neonates in clinical trials find major challenges due to ethical and physiological difficulties; however, dose-finding PK and safety studies are necessary more than ever. The concept of the extrapolation of efficacy data from studies in other populations (e.g., adults) is part of the pediatric study decision tree [132]. As it is reasonable to assume that there is a similar bacteriologic response to those in adults, PK studies, adapted to the unique physiological and maturational characteristics of different neonatal subpopulations (e.g., extremely preterm, preterm, term, those with intrauterine growth restriction), will determine the optimal dose required for targeted exposure and achieve levels similar to adults, although such an approach has its limitations [133]. Undoubtedly, safe and effective use and evaluation in neonates has many challenges [134]. Developmental pharmacology research, which describes the impact of maturation on drug disposition (PK) and drug effects (pharmacodynamics, PD) throughout the neonatal and paediatric age range, is rapidly expanding; drug development needs to incorporate innovative techniques such as preclinical models to study therapeutic strategies, and shift from the sequential enrolment of subgroups, to more rational designs [135].

On the other hand, epidemiological surveillance and the prevention of colonization and infections by MDR-GN bacteria should be priority in every NICU. Strict policies regarding the management of colonized neonates (physical cohorting and staff cohorting), antibiotic stewardship for reducing antibiotic overuse, infection and control practices, and the re-education of staff should be implemented. Key prevention strategies for AMR in neonates target four major pillars: (a) the surveillance of healthcare-associated infections, feedback, and education, (b) the maintenance of skin integrity, (c) the promotion of colonization with normal flora, and (d) the prevention of colonization with pathogens. However, the research on the process of colonization with AMR in neonates and the association with a subsequent infection or other neonatal adverse outcomes has many gaps. Collaboration between NICUs and international networks for the conduction of high-quality studies will help to better understand the circulation and the effect of these pathogens in hospitalized neonates and to find effective tools for their prevention.

## Figures and Tables

**Figure 1 antibiotics-12-01072-f001:**
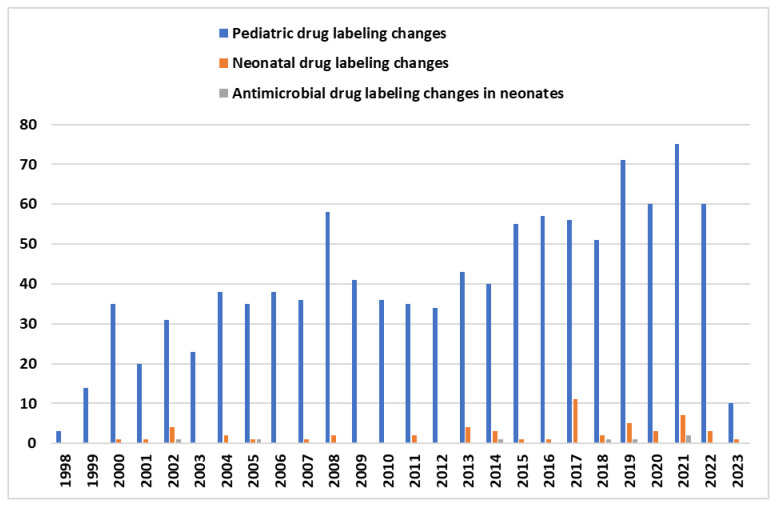
The graph above shows the number of FDA labeling changes of drugs in pediatric population (blue color) since 1998. Orange color depicts all the drug labeling changes for neonates and green color represents antimicrobial drug labeling changes for neonates [23].

**Table 1 antibiotics-12-01072-t001:** PK Parameters of ceftazidime and avibactam in healthy adults and different pediatric ages [34].

Drug/PK Parameter ^a^	Healthy Adults	≥12 to <18 yr	≥6 to <12 yr	≥2 to <6 yr	≥3 m to <2 yr	Neonates to Infants <3 m
Ceftazidime						No PK data An ongoing phase II study (NCT04126031) will provide additional data on CAZ-AVI PK in neonates and young infants with bloodstream infections
C_max_ (mg/liter)	88.1 (14.0)	79.8 (41.8)	81.3 (17.8)	80.1 (14.7)	91.7 (19.6)	
AUC_0-∞_ (h mg/liter)	289 (15.4) ^1^	230.6 (30.7)	221.2 (17.4)	255.32 (43.9) ^2^	286.27 (37.13) ^2^	
t_1/2_ (h) ^b^	3.5 (1.3)	1.7 (0.9–2.8)	1.6 (0.9–1.8)			
Vss (liters)		22.2 (42.0)	13 (17.8)			
CL (liter/h)	7.0 (1.1)	8.7 (45.5)	5.6 (16)			
CL/W (liter/kg/h)		0.16 (37.9)	0.226 (20)			
Avibactam						
C_max_ (mg/liter)	15.2 (14.1)	15.1 (52.4)	14.1 (23)	13.7 (22.4)	16.3 (22.6)	
AUC_0-∞_ (h mg/liter)	42.1 (16) ^1^	36.4 (33.6)	34.8 (22.6)	43.25 (12.14) ^2^	48.99 (10.64) ^2^	
t_1/2_ (h) ^b^	2.3 (0.8)	1.6 (0.9–2.8)	1.7 (0.9-2.0)			
Vss (liters)		31 (53.3)	19.3 (27)			
CL (liter/h)	12 (1.8)	13.7 (52.6)	8.9 (30.2)			
CL/W (liter/kg/h)		0.267 (44.2)	0.359 (35.8)			

^1^ Values are geometric means (percent coefficients of variation) for observed exposures from a phase I study in healthy adult volunteers on day 1 after receiving a single dose of ceftazidime-avibactam (2000 to 500 mg) [35], ^2^ AUC_0-∞_ values are means (SD) based on population pharmacokinetic model predictions. They were comparable across all 4 age groups and in line with adult exposures. ^a^ Values are geometric mean (coefficient of variation [%]), ^b^ Median (range), CL/W: weighted clearance or clearance by body weight, C_max_: maximum drug concentration, AUC_0-∞_: AUC from time zero to infinity, t_½_ half-life, CL: clearance, Vss: volume of distribution at steady state, CV% percentage coefficient of variation, SD: standard deviation.

**Table 2 antibiotics-12-01072-t002:** PK parameters of Ceftolozane–Tazobactam in different pediatric ages and healthy adults [55].

PK Parameters	Group 1 12 to <18y (1.5 g)	Group 2 7 to <12y (18/9 mg/kg)	Group 3 2 to <7y (30/15 mg/kg)	Group 4 3m to <2y (30/15 mg/kg)	Group 5, GA > 32w PNA = 7d to <3m (20/10 mg/kg)	Group 6, GA ≤ 32w PNA = 7d to <3m (20/10 mg/kg)	Adults (1.5 g)
Ceftolozane							
AUC_0-∞_	133 (104–171)	107 (85.7–135)	186 (135–255)	202 (158–259)	164 (131–205)	137 (99.6–189)	172 (13.8)
Cmax	63.5 (50.2–80.4)	56.2 (45.3–69.7)	96.6 (71.2–131)	96.6 (71.2–131)	45.0 (36.3–55.9)	45.2 (33.3–61.2)	69.1 (11.3)
t1/2	1.45 (16.7)	1.29 (9.6)	1.48 (35.5)	1.63 (69.0)	2.21 (37.6)	3.14 (0.9)	2.77 (30.0)
CL	0.146 (27.0)	0.168 (21.3)	0.162 (31.1)	0.149 (43.2)	0.118 (36.0)	0.147 (6.8)	0.837
Vss	0.274 (25.7)	0.296 (22.0)	0.312 (19.5)	0.340 (21.1)	0.394 (12.6)	0.388 (26.9)	0.209
Tazobactam							
AUC_0-∞_	17.5 (12.6–24.2)	10.2 (6.68–15.5)	28.9 (19.0–43.9)	29.9 (21.6–41.4)	24.9 (18.0–34.4)	22.3 (14.7–34.0)	24.4 (17.9)
Cmax	14.0 (8.59–22.9)	9.25 (5.92–14.5)	24.8 (13.2–46.6)	22.4 (13.8–36.6)	11.7 (7.48–18.3)	12.1 (6.43–22.7)	18.4 (15.9)
t1/2	0.702 (38.7)	0.544 (3.1)	0.770 (34.2)	0.815 (85.1)	1.09 (32.0)	0.875 (20.4)	0.91 (26.2)
CL	0.556 (53.9)	0.886 (23.1)	0.519 (44.8)	0.502 (34.7)	0.385 (34.1)	0.452 (24.9)	0.294
Vss	0.474 (69.6)	0.740 (30.2)	0.513 (49.2)	0.574 (36.2)	0.668 (19.8)	0.667 (29.3)	0.259

GA: gestational age, PNA: postnatal age, y: years, m: months. Statistics for AUC_0-∞_ and Cmax: Geometric least-squares mean and confidence interval. Statistics for Tmax: Median and range (minimum, maximum). Statistics for t1/2, CL and Vss: Geometric mean and percent geometric coefficient of variation, CV% = 100 × sqrt (exp(s2) − 1), where s2 is the observed between-patient variance on the natural log-scale.

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
