# Peer review of "Use of Newer and Repurposed Antibiotics against Gram-Negative Bacteria in Neonates"

_antibiotics, 2023, doi:10.3390/antibiotics12061072_

Round 1

Reviewer 1 Report

Kontou et.al, summarized data about some novel or repurposed antibiotics that are active against MDR Gram-negative (MDR-GN) bacteria.

Comments:

1.       Can you please include a paragraph about physicians/doctors or clinical trials have calculated doses for neonates and pre-terms? Have they used any PBPK modeling? If yes, how have the predictions of dose and hence the PK parameters of c-t profiles matched the observed data

2.       Can you please include a paragraph about drug-drug interactions (DDI)? The article mentions of often there is co-medications. Is there any data on DDI available in pediatric patients?

3.       Page 3 line 133 use the word ‘low’ instead of ‘poor’ for plasma protein binding.

4.       Is it possible to summarize the PK parameters (CL, Vss, AUC, dose, half-life etc. ) for adults and different pediatric populations (pre-terms, neonates etc) of the drugs mentioned in the article. If available, perhaps a table. This will help readers to see the differences in adult and pediatric population

5.       Is it possible to draw a graph of number of drugs approved in last couple decades for infectious diseases? This will give the reader a perspective of where the field is moving in terms of drug approvals

6.       If possible, can a table be made of off-label use, side-effects of all the drugs mentioned in this article and perhaps some other drugs.

Author Response

Please see the attachmenτ.

Reviewer 2 Report

Great article! Very glad to recommend for publishing!

Reviewer 3 Report

This manuscript gives a comprehensive review of newer/novel antibiotic against Gram-negative bacteria in neonates. It is well-written. I have some comments as follows:

1.     L 53-61: The development of carbapenem resistance is complex. Overproduction of AmpC or other ESBL may cause the carbapenem resistance phenotype. (PMID: 32494169, 27799202) Suggest also add this information to the text.

2.     Some typing errors need to be checked and corrected. For example: line 117.

3.     L 210-219: Besides resistance to ceftazidime/avibactam, the rapid identification of intrinsic resistance mechanisms such as Ambler class B carbapenemase is crucial for appropriate antibiotic selection. Point-of-care testing and regional epidemiology data are of great value.

4.     L 255-257: This context is repeated noted in previous paragraphs. Suggest delete or modify it.

5.     L 337-338: What is the meaning of %T>MIC? Time above MIC?

6.     Cefiderocol is an important drug to treat multidrug resistance microorganisms. There are also some reports in children. Suggest also make some descriptions about this novel antibiotic.

7.     L 467: Some abbreviations, such as MRSA, were already mentioned in L 41.

8.     L 521-523: How to draw this conclusion?

The manuscript is well-written.
